# Fracture and Size Effect of PFRC Specimens Simulated by Using a Trilinear Softening Diagram: A Predictive Approach

**DOI:** 10.3390/ma14143795

**Published:** 2021-07-07

**Authors:** Fernando Suárez, Jaime C. Gálvez, Marcos G. Alberti, Alejandro Enfedaque

**Affiliations:** 1Departamento de Ingeniería Mecánica y Minera, Universidad de Jaén, 23071 Jaén, Spain; fsuarez@ujaen.es; 2Departamento de Ingeniería Civil-Construcción, Universidad Politécnica de Madrid, E.T.S.I. Caminos, Canales y Puertos, 28040 Madrid, Spain; marcos.garcia@upm.es (M.G.A.); alejandro.enfedaque@upm.es (A.E.)

**Keywords:** size effect, polyolefin-fibre-reinforced concrete, trilinear softening function, cohesive model

## Abstract

The size effect on plain concrete specimens is well known and can be correctly captured when performing numerical simulations by using a well characterised softening function. Nevertheless, in the case of polyolefin-fibre-reinforced concrete (PFRC), this is not directly applicable, since using only diagram cannot capture the material behaviour on elements with different sizes due to dependence of the orientation factor of the fibres with the size of the specimen. In previous works, the use of a trilinear softening diagram proved to be very convenient for reproducing fracture of polyolefin-fibre-reinforced concrete elements, but only if it is previously adapted for each specimen size. In this work, a predictive methodology is used to reproduce fracture of polyolefin-fibre-reinforced concrete specimens of different sizes under three-point bending. Fracture is reproduced by means of a well-known embedded cohesive model, with a trilinear softening function that is defined specifically for each specimen size. The fundamental points of these softening functions are defined a priori by using empirical expressions proposed in past works, based on an extensive experimental background. Therefore, the numerical results are obtained in a predictive manner and then compared with a previous experimental campaign in which PFRC notched specimens of different sizes were tested with a three-point bending test setup, showing that this approach properly captures the size effect, although some values of the fundamental points in the trilinear diagram could be defined more accurately.

## 1. Introduction

In the case of plastic limit analysis or elasticity assessed up to a strength limit, results are independent of the specimen size, but in the case of elements made of quasi-brittle materials such as concrete that are evaluated beyond the proportionality limit, the nominal strength is dependent on the specimen size.

Size effect on plain concrete is well known and is the reason why fracture develops at lower values of the nominal strength when the size of a concrete specimen increases while keeping the same proportions [1]. When fracture in plain concrete is numerically reproduced, the size effect can be correctly captured by means of a cohesive zone formulation that uses a well-characterised softening diagram [2,3,4]. In fact, as Bažant states, the cohesive crack model proposed by Hillerborg can be considered as the most realistic among simple models when quasi-brittle fractures are studied [5].

The use of fibres as reinforcement in concrete has been studied for decades and has been traditionally developed by using steel fibres [6,7] but has been boosted in recent years, and the range of fibres used for this purpose has increased [8,9,10,11], with polyolefin fibres being one of the most recent types. Polyolefin-fibre-reinforced concrete (PFRC) is experiencing great development in recent years, due to its good mechanical behaviour and the fact that it reduces and, in some cases, even eliminates some of the problems observed in steel-fibre-reinforced concrete (SFRC) such as corrosion, sensitivity to magnetic fields, or wear and tear of machinery related to its production (concrete pumps and mixers, for example), making PFRC particularly suitable for some uses. The effect of these fibres on the properties of PFRC has been studied in depth during the last years for traditional vibrated concrete [12], self-compacting concrete [13], and in combination with steel fibres [14]. Many aspects of PFRC are already studied, such as the fibre distribution depending on the production process [15] or how it affects fracture in mode I [16] and mode II [17]. Although this material is starting to count with initial examples of use as a structural material [18,19], one of the reasons why PFRC is still not becoming as widespread as it could is probably the scarce experience with it and the uncertainty on its behaviour in real engineering works under certain situations. One of the aspects that must be clarified is the size effect; this is of paramount importance if the material properties measured at a laboratory scale are to be used for designing larger structures.

There is not much information about the size effect in fibre-reinforced concrete (FRC), especially in the case of PFRC, given that it is a relatively recent material. In the case of SFRC, some studies can be found [20,21], and in the case of PFRC, an experimental campaign has been recently carried out [22], which has shown that the nominal strength at the limit of proportionality is governed by the matrix (concrete), and the post-cracking residual strength is governed by the fibres.

In previous works, the use of a cohesive zone formulation fed with a trilinear softening curve has proven to be very convenient for reproducing the fracture process in FRC [23], but it must be adapted depending on several factors such as the fibre length, the fibre proportion [16], and the specimen size [24]. This trilinear softening diagram describes the contribution of matrix and fibres in the fracture process which, due to the different elastic moduli of both materials, begin to significantly work at different stages of load transmission. Considering the trilinear diagram shown in Figure 1, the initial point *t* identifies the fracture of the concrete matrix, *k* the point at which the contribution of fibres starts to predominate over the contribution of the matrix, *r* the maximum remanent contribution of fibres, and *f* the eventual failure of the material.

In [16], some parameters of the PFRC mix were identified, and some expressions were also proposed to define the fundamental points of the trilinear diagram (*k* and *r* points). In addition to this, in [25] the length and orientation of fibres were observed as key parameters to define the trilinear diagram, also identifying a higher threshold of the PFRC behaviour obtained testing specimens with long fibres oriented in the optimum direction.

From the numerical point of view, there exist many approaches and models that help to simulate fracture. In many cases, these models are calibrated using the experimental results of the test simulated, but this does not guarantee that the parameters represent any other case different from the one under study. From this point of view, the most interesting approach consists of finding models that can reproduce fracture in a predictive way, that is, a model that is fed with parameters obtained by experimental tests that are different from the loading case that wants to be simulated. This type of model is considered less biased and more representative of the material than a specific loading case.

The main aim of this contribution is to reproduce fractures on different size specimens of PFRC using a predictive approach. A cohesive model and a softening diagram that corresponds to a trilinear function defined a priori was employed. Using the knowledge obtained in previous works, the coordinates of each of the fundamental points *t*, *k*, *r* and *f* were identified. To do this, the experimental results of [22] were reproduced and compared through a finite element analysis by using an embedded cohesive crack formulation. In the following sections, the experimental work used as a reference of the size effect in PFRC is briefly described, then the main features of the embedded cohesive crack model used to numerically reproduce fracture are presented, and the trilinear softening functions used with each specimen size are obtained by means of the expressions proposed in [16]. Since, as will be later discussed, some parameters of the diagram (more specifically, $w_{r}$ and $w_{f}$) are estimated, some parts of the load–displacement curves obtained numerically only agree partially with the experimental results. Therefore, in the final part of this paper, an analysis of the influence of such parameters is carried out, and some conclusions are highlighted.

## 2. Experimental Benchmark

In order to compare the numerical simulations with experimental results, the campaign described in [22] was used. For a detailed description of this campaign, the reader is encouraged to read the referenced work, since here only the main aspects relevant for the present study are presented. Table 1 shows the concrete composition used in this campaign, which corresponds to a self-compacting concrete with 10 kg of fibres per m${}^{3}$ (SCC10).

Concrete reinforcement consists of 48 mm long polyolefin macrofibres with an embossed surface. The main properties of these fibres can be consulted in Table 2. More information on these fibres can be found in [12].

The experimental campaign of reference involved three-point bending tests carried out on three samples of each size, following the guidelines of the EN-14651 standard [26] (except for the specimen sizes and notch dimensions). Figure 2 shows a schematic drawing of the experimental setup, and Table 3 shows the dimensions of the specimens. In all cases, the concrete composition was the same, and 48 mm long polyolefin fibres were used in a proportion of 10 kg/m${}^{3}$. The scheme on the left of Figure 2 shows the proportions of the specimens, which remained equal for each size, and the scheme on the right shows the specimen at an intermediate state of the test, when the cracking process was in progress and propagated vertically from the notch tip.

In these tests, the load and the displacement were obtained from the testing machine, and the evolution of the crack mouth opening displacement (CMOD) was measured by means of a digital image correlation system (DIC). To compare the experimental results, two main diagrams were employed: load versus displacement of the application point of the load and load versus CMOD. Figure 2 shows these values in the scheme of a damaged specimen during the test.

## 3. Embedded Cohesive Crack Model

The crack process is modelled by using the finite element analysis and adapting a formulation based on the cohesive zone approach developed by Hillerborg [27], inspired by the work of Dugdale [28] and Barenblatt [29]. This formulation simulates fracture inside an element using the strong discontinuity approach and was initially developed for concrete [30,31] but later adapted to brickwork masonry elements [32] and fibre-reinforced cementitious materials [16,23,24].

The cohesive zone approach relies on the experimental evidence that fracture usually develops under a predominant local mode I. Thus, this approach assumes that the cohesive stress vector t is perpendicular to the crack opening and parallel to the crack displacement vector w, which is expressed by (Equation 1).
(1)$$t=\frac{f(\tilde{w})}{\tilde{w}}w\;\;\mathrm{with}\;\tilde{w}=\max\left(\left|w\right|\right)$$
where $f(\left|\tilde{w}\right|)$ stands for the material softening function, defined in terms of an equivalent crack opening $\tilde{w}$. This equivalent crack opening stores the maximum historical crack opening to account for possible unloading scenarios. In this case, the softening diagram is defined as trilinear, as shown in Figure 3, and the load–unload branches follow lines towards the origin in all cases. The trilinear diagram is defined by the following expression: (2)$$\sigma =\left\{\begin{array}{cc} f_{ct}+\left(\frac{\sigma_{k}-f_{ct}}{w_{k}}\right)\cdot w & \mathrm{if}\;0<w\leq w_{k}\\ \sigma_{k}+\left(\frac{\sigma_{r}-\sigma_{k}}{w_{r}-w_{k}}\right)\cdot \left(w-w_{k}\right) & \mathrm{if}\;w_{k}<w\leq w_{r}\\ \sigma_{r}+\left(\frac{-\sigma_{r}}{w_{f}-w_{r}}\right)\cdot \left(w-w_{r}\right) & \mathrm{if}\;w_{r}<w\leq w_{f}\\ 0 & \mathrm{if}\;w>w_{f} \end{array}\right.$$

In the finite element models presented later, the embedded cohesive crack formulation is used with constant strain triangular elements. Cracking can only develop in three directions, each parallel to the element sides and at mid height, which guarantees that local and global equilibria are satisfied. Figure 4 shows the only three possible crack paths in an element.

Once the crack direction is defined, the element is divided into two parts, A${}^{+}$ and A${}^{-}$, and the stress vector t is constant along the crack, expressed by (Equation 3).
(3)$$t=\frac{A}{hL}\sigma \cdot n$$
where *A* stands for the area of the element, *h* for the height of the triangle over the side opposite to the solitary node, *L* for the crack length in the element, and n for the unit vector normal to that side and to the crack. Since the crack is parallel to one side of the triangular element and is placed at mid height, Expression (Equation 3) turns into t=σ·n (the reader can find more details of this and other aspects of the model in [30]).

The material outside the crack is assumed to be elastic, and the crack displacement vector w is solved considering that the stress tensor can be obtained by subtracting an inelastic part, which considers the contribution of the crack displacement to the elastic prediction computed using the apparent strain by means of (Equation 4).
(4)$$\sigma =E:\left[\epsilon^{a}-{\left(b^{+}⊗w\right)}^{S}\right]\cdot n$$
where E is the elastic tangent tensor, $\epsilon^{a}$ the apparent strain vector obtained with the nodal displacements, $b^{+}$ the gradient vector of the shape function that corresponds to the solitary node, which can be easily obtained in this case by (Equation 5), superscript *S* indicates the symmetric part of the resulting tensor: the double-dot product (${\left(A:b\right)}_{ij}=A_{ijkl}b_{kl}$), and ⊗ the direct product (${\left(a⊗b\right)}_{ij}=a_{i}b_{j}$).
(5)$$b^{+}=\frac{1}{h}n$$

Since the stress vector t can be obtained as t=σ·n, using the expression of σ obtained with (Equation 4) and the expression of t in terms of the crack opening (Equation 1), the following expression is defined:$$\frac{f(\tilde{w})}{\tilde{w}}w=\left[E:\epsilon^{a}\right]\cdot n-\left[E:{\left(b^{+}⊗w\right)}^{S}\right]\cdot n$$
which can be rewritten as
(6)$$\left[\frac{f(\tilde{w})}{\tilde{w}}1+n\cdot E\cdot b^{+}\right]\cdot w=\left[E:\epsilon^{a}\right]\cdot n$$
where 1 stands for the second-order identity tensor. Using an iterative process (such as the Newton–Raphson method), the crack displacement *w* that satisfies (Equation 6) can be obtained.

This model is implemented using a UMAT subroutine in ABAQUS and, since vectors n, $b^{+}$, crack length *L*, and the element area *A* are computed using the nodal coordinates for each element, it reads an external file with this information.

## 4. Definition of the Trilinear Softening Diagrams

As observed in [16], there are several parameters that can be experimentally measured and help to define the trilinear diagram for the PFRC. Apart from the fracture parameters of plain concrete ($G_{F}$ and $f_{t}$), which define the first part of the diagram, these parameters are the volume of fibres ($V_{f}$), the orientation factor (θ) and the percentage of pulled out fibres at the fracture surface (%Pulled−out). With the help of $V_{f}$, the angle ϕ can be obtained by means of (Equation 7).
(7)$$\phi =-3.6046+5.0625\cdot \left(1-e^{\left(-6.55\cdot V_{f}\right)}\right)$$

This angle serves to identify the second point of the diagram (point *k*), which is the intersection of the softening function of plain concrete (here considered as an exponential function: $\sigma =f_{t}\cdot \exp\left(-\frac{f_{t}\cdot w}{G_{F}}\right)$) with a line passing through the origin with a direction defined by ϕ (see Figure 5).

By using the three main parameters mentioned before and the ultimate tensile strength of the fibres ($\sigma_{u}$), the maximum remaining strength ($\sigma_{r}$) can be obtained with (Equation 8).
(8)$$\sigma_{r}=\left(1-\%Pulled-out\right)\cdot V_{f}\cdot \theta \cdot \sigma_{u}$$

Considering the scheme of the trilinear diagram shown in Figure 1, the first two points can be identified as follows: point *t* is identified by $f_{t}$, which can be experimentally obtained, while point *k* depends on the volume fraction of fibres ($V_{f}$) by means of the ϕ angle defined with (Equation 7) and the softening function of plain concrete. Table 4 shows the intermediate values that result of this calculation.

As regards the remaining two points, *r* and *f*, the value of $\sigma_{r}$ can be obtained with (Equation 8). Table 5 shows the results of this calculation for each size, and $\sigma_{f}$ is, obviously, equal to 0, but $w_{r}$ and $w_{f}$ must be estimated; they depend on the fibre length, but there are no specific expressions to obtain them. In this case, $w_{r}$ is estimated as equal to 1.65 mm, since this was the value adopted in [24] for simulating fracture in specimens made with 48 mm long fibres of the same kind as those used here. As regards $w_{f}$, this value is related to the maximum crack opening before completely losing the bonding between the fibres and the matrix; therefore, it is assumed to be proportional to the fibre length. Thus, since in [16] specimens made with 60 mm long fibres were modelled using $w_{f}=7.5$ mm, here, a value of $w_{f}=\frac{48}{60}\cdot 7.5=6.0$ mm is adopted. Figure 6 shows the resulting trilinear softening diagrams for all three sizes.

## 5. Results and Discussion

Fracture of the three specimen sizes analysed in [12] was carried out using the finite element method, and a displacement control was used to drive the fracture evolution with good convergence. The simulations were computed using ABAQUS [33], and the fracture was reproduced by means of a UMAT subroutine that implemented the previously described material behaviour.

Figure 7 shows the three meshes used in this work with the same scale. In all three meshes, the region connecting the notch tip with the load application point was refined in order to better capture the fracture process, while the rest of the specimen was meshed with larger elements, which helped to notably reduce the time of computation. The models were formed by a number of nodes smaller than 800 and a number of triangular finite elements smaller than 1500, thus keeping the model size small enough to have models that perform efficiently. These simulations were run on a computer with an Intel Xeon E5-1620 processor with 4 cores at 3.5 GHz, although only one was used since the user subroutine that reproduces the material behaviour does not allow parallel computing; all the simulations took around 150 min to run. In the case of the large size model (L), the side of minimum element size was around 7 mm, in the case of the medium size model (M), 3.5 mm, and in the case of the small size model (S), 2 mm. The refinement of these meshes was designed based on previous works (see [24]), in which the mesh dependence was already analysed.

Figure 8 shows the load–load displacement and load–CMOD diagrams for all three sizes and compares them with the experimental results. Each specimen size is identified by a different colour: red for large size, blue for medium size, and green for small size. The shades behind the diagrams correspond to the experimental envelopes, with the same colour code used in the diagrams; therefore, the red shade corresponds to the experimental envelope of the large specimens, the blue shade to the experimental envelope of medium specimens, and the green shade to the experimental envelope of small specimens. Apart from the overestimation of the initial peak, which is a known issue when this type of numerical modelling is used, especially in large-sized specimens [34], the models reproduce the experimental results reasonably well. This agreement is particularly good, in the case of the medium size, and presents some differences in the last part of the load–load displacement diagrams, in the cases of large and small sizes, in which the numerical model tends to underestimate the specimen’s remaining strength.

It is also worth noting that in the case of the experimental results, the maximum remanent load occurs at a larger load displacement if compared with the medium and large sizes, while in the case of the numerical results, this maximum load after the first peak occurs approximately at the same load displacement and, in all cases, following a very linear trend. These trends are depicted by dashed lines on the load–load displacement diagrams of Figure 8.

These results show that expressions (Equation 7) and (Equation 8), defined in the past by analysing the fracture behaviour of different PFRC mixes, well describe the general behaviour of this material and take into account the main parameters: the volume of fibres in the mix ($V_{f}$), the orientation of fibres with respect to the fracture surface (θ), and the quality of bonding between the fibres and concrete, expressed by the fraction of fibres that are pulled out at the fracture surface (%Pulled−out). Nevertheless, the parameters used to define the trilinear softening diagrams, abscissa values of points *k* and *f*, are only estimated based on previous experiences with this type of model, but there are no expressions proposed for them yet. In the following section, the influence of these two values, $w_{r}$ and $w_{f}$, is studied to understand how they modify the diagrams, which can help to propose expressions to quantify them.

## 6. Study on the Influence of $w_{r}$ and $w_{f}$

### 6.1. Influence of $w_{r}$

To understand how the value of $w_{r}$ modifies the load–load displacement and load–CMOD diagrams for each specimen size, the trilinear diagrams of Table 6 were considered as a reference, and new ones were defined by modifying the value of $w_{r}$, keeping the rest unchanged. Figure 9 shows these trilinear functions and Figure 10 the resulting load–load displacement and load–CMOD diagrams, considering the reference value of $w_{r}$, 1.65 mm, and two alternative values, 1.85 mm and 2.05 mm, were identified by dashed lines and dotted lines, respectively; these values have been adopted as reasonable alternatives of the reference value according to previous experiences with this type of simulation. As described for Figure 8, red, blue and green shades identify the experimental envelopes of large, medium, and small specimens, respectively.

As a consequence of defining larger values of $w_{r}$, the maximum remanent load is reached later in the test, as could be expected, but it is also interesting to observe that the maximum load becomes slightly lower. This makes the simulations of large and medium sizes agree somewhat better with the experimental envelope. The maximum remanent load still shows proportionality with the specimen size and the experimental observation is not reproduced, since, in the small size case, it still occurs earlier than in the experimental tests.

These results suggest that the value of $w_{r}$ has a strong influence on the diagram, i.e., on the load displacement at which the maximum remanent load occurs, as well as on its value, since the larger $w_{r}$ is, the smaller the maximum remanent load becomes. This is due to the uneven stress evolution along the fracture plane during the test; if $w_{r}$ is smaller, the slope of the *k*–*r* line (see Figure 3) is steeper; therefore, the lower fibres of the fracture plane, which are the first to be damaged and the most relevant from the point of view of the bearing capacity of the section due to their position, begin to recover their loading capacity earlier. At this point, as there is a smaller fraction of the ligament damaged, the sample is capable of reaching a higher value of the remanent load before its final decay. On the contrary, if $w_{r}$ is higher, the loading capacity of the lower fibres of the bearing section is recovered later, and therefore, a larger portion of the section is damaged, which leads to a smaller value of the maximum remanent load.

### 6.2. Influence of $w_{f}$

To analyse how $w_{f}$ affects the fracture behaviour in the simulations of each size, the same strategy as with $w_{r}$ was followed. Using the trilinear diagrams of Table 6 as the reference, two more diagrams were created for each size by modifying only the abscissa of point *f*, using values considered as reasonable based on previous experiences with these simulations. Figure 11 shows these trilinear diagrams and Figure 12 the curves obtained with them in the numerical simulations. As in Figure 8 and Figure 10, red, blue, and green shades identify the experimental envelopes for each specimen size.

As expected, these diagrams show that when $w_{f}$ is larger, the load decay in the last part of the test is delayed, which makes the numerical simulations more akin to the experimental observations in the case of the large size, maintaining a good agreement with the other two sizes. Although the behaviour of the model is very similar up to the maximum remanent load, there are small differences among each of the three values of $w_{f}$ used in each specimen size. Taking a closer look at the diagrams around the region where the maximum remanent load occurs, it can be observed that this occurs at a slightly higher value of the load displacement and the value of this load is slightly higher, although the differences are almost negligible. This behaviour is expected, since once the damage starts in the lower fibres of the ligament and progresses beyond the *r* point, their bearing capacity decays at a slower rate as the crack opens. Since the maximum remanent load is the result of a competition between the load lost by the more damaged fibres that have already reached the *r* point and the increased bearing capacity of less damaged fibres that have still not reached the *r* point, a softer slope of the *r*–*f* line in the trilinear diagram results in a higher remanent load that occurs later in the test (see Figure 3).

Lastly, regarding the experimental trend of the maximum remanent load observed experimentally, here, again, it presents a proportional evolution with the specimen size. As observed with the alternative softening diagrams obtained by modifying $w_{r}$, the maximum remanent load occurs approximately at the same load displacement for all three sizes, while the experimental results (see the load–load displacement diagram in Figure 8) show a larger value for the small size; therefore, this aspect is not captured by the numerical simulations.

As the previous results show, modifying $w_{r}$ or $w_{f}$ has effect on the maximum remanent load and the final decay branch of the diagrams, but the alternative trilinear softening diagrams used suggest that these values also depend on the specimen size. Estimating $w_{r}$ and $w_{f}$ as fixed values for all specimen sizes results in a proportional evolution of the maximum remanent load that follows a linear trend in the load–displacement diagrams, which does not agree with the experimental observations. The work carried out in past papers provide a good tool to reproduce the behaviour of PFRC elements in a predictive way, capturing reasonably well the size effect of this type of materials, but the results shown here suggest that expressions such as those used to define $\sigma_{k}$, $w_{k}$ and $\sigma_{r}$ (see Equations (Equation 7) and (Equation 8)) should be found for $w_{r}$ and $w_{f}$ in order to have a tool that captures the whole fracture behaviour of this composite material and be fully predictive.

## 7. Conclusions

In this work, the numerical modelling of the size effect by means of a cohesive model fed with a trilinear softening function was studied using a predictive method. A three-point bending test on specimens of three sizes was numerically reproduced and compared with experimental data from previous works. The trilinear diagrams for each size were defined by expressions obtained in previous experimental campaigns, resulting in good agreement with the lab observations.

From the work presented above, the following conclusions can be drawn:The complete fracture behaviour of PFRC specimens can be numerically simulated using a predictive trilinear cohesive crack model, which can be defined a priori by means of empirical expressions obtained with lab tests different from those simulated. This diagram is defined by four points, with coordinates that depend on PFRC mechanical characteristics, i.e., the tensile strength of the matrix, the proportion of fibres, and the orientation factor. Abscissa values $w_{r}$ and $w_{f}$ (see Figure 3) are fixed based on experimental results obtained in previous literature. It is still an unsolved challenge to obtain expressions to estimate $w_{r}$ and $w_{f}$ using the mechanical characteristics of the PFRC.The softening diagrams are not equal for all specimen sizes and should be adjusted for each of them. This is mainly due to a different orientation factor that varies with the size of the specimen.The maximum remanent loads obtained for each size present a linear trend on the load–displacement diagram, which does not agree completely with the experimental observations, although the load–displacement and load–CMOD curves properly agree with the experimental envelopes for the three studied sizes.Modifying $w_{r}$ and $w_{f}$ affects the maximum remanent load on the load–displacement diagram and modifies the last part of this diagram but cannot capture the nonlinear trend of the remanent load among specimen sizes.

## Figures and Tables

**Figure 1 materials-14-03795-f001:**
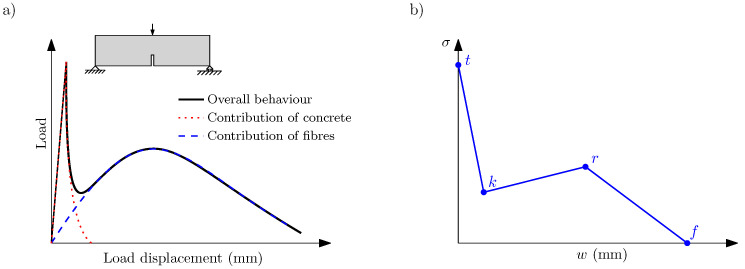
(**a**) Load–displacement diagram obtained in a three-point bending test with a PFRC specimen; (**b**) trilinear softening diagram.

**Figure 2 materials-14-03795-f002:**
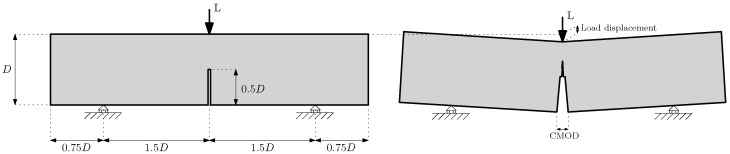
**Left**: scheme of a three-point bending test and specimen geometry; **right**: scheme of crack propagation from the notch tip during the test.

**Figure 3 materials-14-03795-f003:**
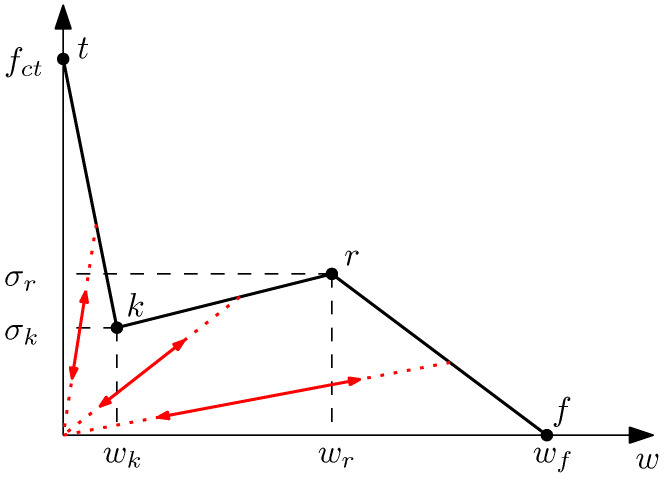
Scheme of a trilinear softening function. Load–unload branches follow a line towards the origin.

**Figure 4 materials-14-03795-f004:**
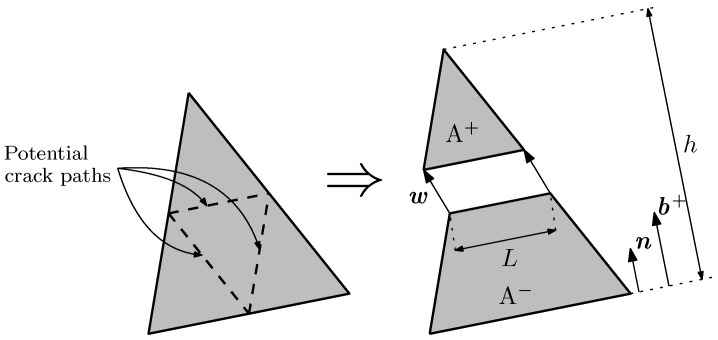
Potential crack paths (**left**) and geometrical definitions of w, n, and $b^{+}$ (**right**).

**Figure 5 materials-14-03795-f005:**
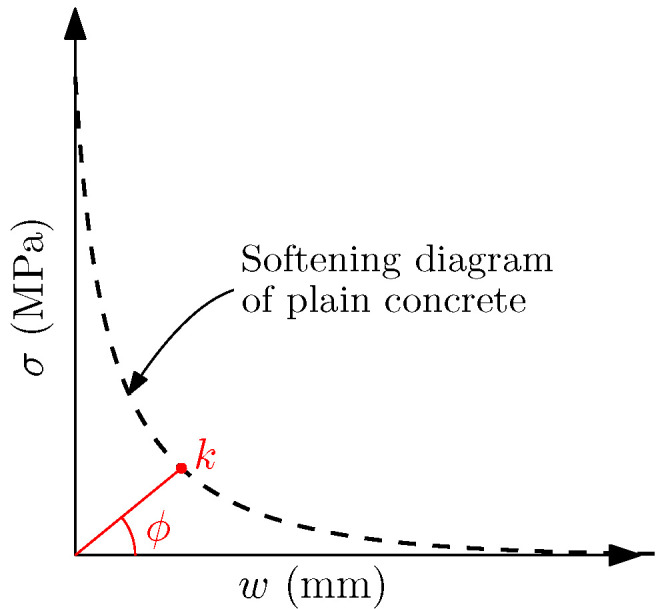
Identification of *k* point of the trilinear diagram by means of the angle ϕ.

**Figure 6 materials-14-03795-f006:**
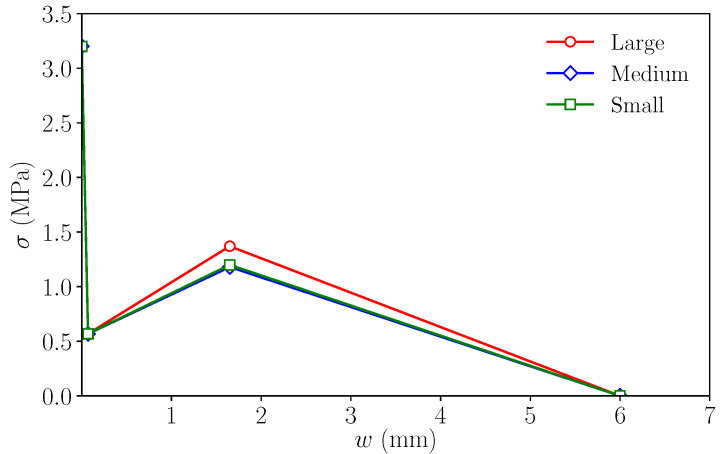
Initial trilinear softening diagrams.

**Figure 7 materials-14-03795-f007:**
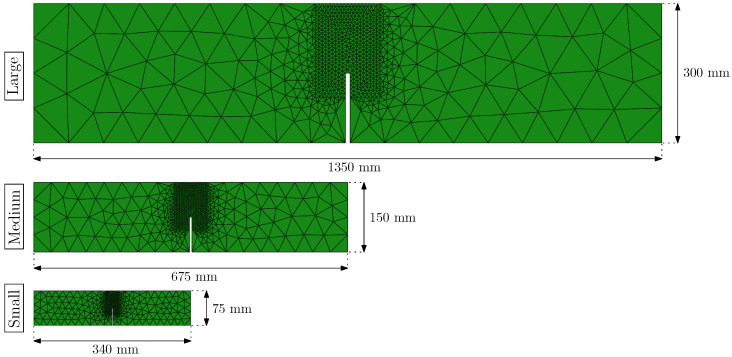
FEM meshes used in the simulations for each specimen size.

**Figure 8 materials-14-03795-f008:**
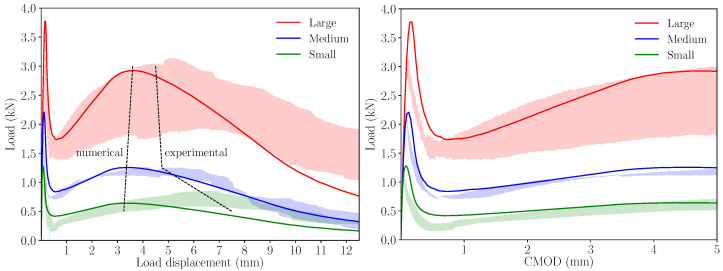
Numerical results compared with the experimental envelopes; each specimen size is identified by a different colour. Experimental envelopes correspond to three specimens tested.

**Figure 9 materials-14-03795-f009:**
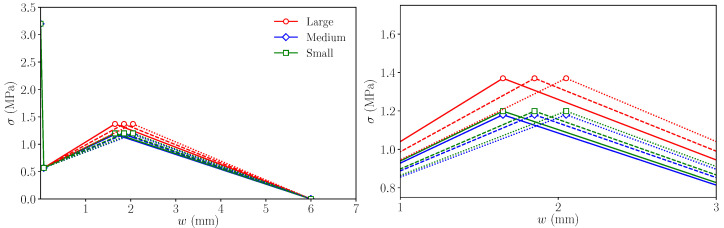
**Left**: trilinear diagrams used to study the influence of $w_{r}$ on the numerical simulations; **right**: detail of the diagrams around point *r* (see Figure 3). Continuous lines represent the trilinear functions with $w_{r}$ = 1.65, dashed lines the functions with $w_{r}$ = 1.85, and dotted lines the function with $w_{r}$ = 2.05.

**Figure 10 materials-14-03795-f010:**
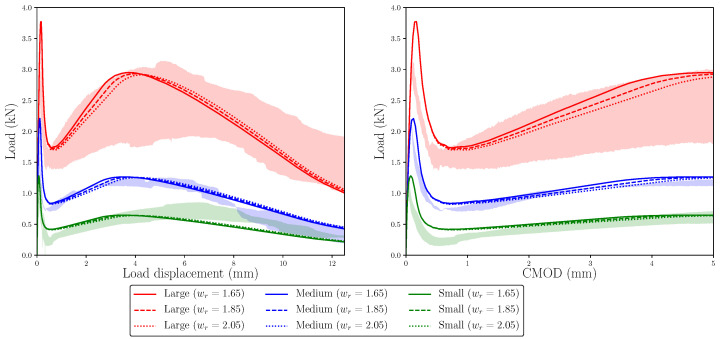
Numerical results compared with the experimental envelopes to study the influence of $w_{r}$ on the numerical simulations. The trilinear functions used in each case are shown in Figure 9. Experimental envelopes correspond to three specimens tested.

**Figure 11 materials-14-03795-f011:**
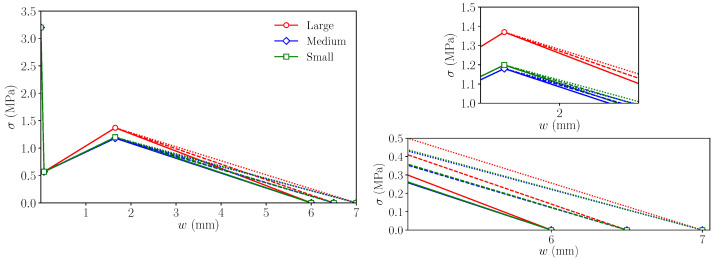
Left: trilinear diagrams used to study the influence of $w_{f}$ on the numerical simulations; upper right: detail of the diagrams around point *r*; lower right: detail of the diagrams around point *f* (see Figure 3). Continuous lines represent the trilinear functions with $w_{f}$ = 6.0, dashed lines, the functions with $w_{f}$ = 6.5, and dotted lines, the function with $w_{f}$ = 7.0.

**Figure 12 materials-14-03795-f012:**
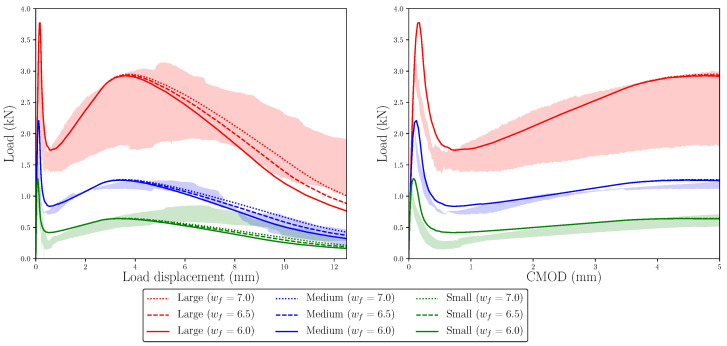
Numerical results compared with the experimental envelopes to study the influence of $w_{f}$ on the numerical simulations. The trilinear functions used in each case are shown in Figure 11. Experimental envelopes correspond to three specimens tested.

**Table 1 materials-14-03795-t001:** Concrete composition.

Material	SCC10
Cement (kg/m${}^{3}$)	375
Limestone (kg/m${}^{3}$)	200
Water (kg/m${}^{3}$)	188
w/c	0.5
Gravel (kg/m${}^{3}$)	245
Grit (kg/m${}^{3}$)	367
Sand (kg/m${}^{3}$)	918
Superplasticizer (% cement)	1.25
PF48 (kg/m${}^{3}$)	10

**Table 2 materials-14-03795-t002:**
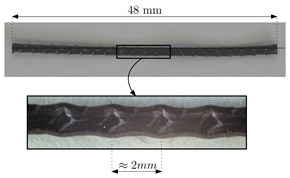
Fibres properties.

Material density (g/cm${}^{3}$)	0.910
Eq. diameter (mm)	0.903
Tensile strength (MPa)	>500
Modulus of elasticity (GPa)	>9

**Table 3 materials-14-03795-t003:** Specimens’ dimensions.

Specimen	Length (mm)	Width (mm)	Height (mm)	Notch (mm)
Large	1350	50	300	150
Medium	675	50	150	75
Small	340	50	75	37.5

**Table 4 materials-14-03795-t004:** Intermediate values for obtaining point *k* of the trilinear diagram.

	$f_{t}$ (MPa)	$G_{F}$ (N/mm)	ϕ	$w_{k}$ (mm)	$\sigma_{k}$ (MPa)
Small/Medium/Large	3.2	0.13	1.448	0.07143	0.57715

**Table 5 materials-14-03795-t005:** Intermediate values for obtaining $\sigma_{r}$ for all three considered sizes.

	θ	%Pulled−Out	$V_{f}$	$\sigma_{u}$ (MPa)	$\sigma_{r}$ (MPa)
Small	0.63	0.54	0.011	376	1.20
Medium	0.62	0.54	0.011	376	1.18
Large	0.72	0.54	0.011	376	1.37

**Table 6 materials-14-03795-t006:** Parameters of the softening diagrams for each specimen size.

	Small	Medium	Large
$w_{t}$ (mm)	0.00	0.00	0.00
$\sigma_{t}$ (MPa)	3.20	3.20	3.20
$w_{k}$ (mm)	0.07	0.07	0.07
$\sigma_{k}$ (MPa)	0.57	0.57	0.57
$w_{r}$ (mm)	1.650	1.650	1.650
$\sigma_{r}$ (MPa)	1.20	1.18	1.37
$w_{f}$ (mm)	6.00	6.00	6.00
$\sigma_{f}$ (MPa)	0.00	0.00	0.00

## Data Availability

The data presented in this study are available from the authors on request.

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
