# Peer review of "Fracture and Size Effect of PFRC Specimens Simulated by Using a Trilinear Softening Diagram: A Predictive Approach"

_materials, 2021, doi:10.3390/ma14143795_

Round 1
Reviewer 1 Report
In this study, a predictive methodology is used to reproduce fracture of polyolefin fiber reinforced concrete specimens of different sizes under three-point bending. Overall speaking, this is a very interesting paper. The numerical method is well explained and the experiments are well designed. The conclusions are also well supported. I think this is paper is ready for publication.
Author Response
In this study, a predictive methodology is used to reproduce fracture of polyolefin fiber reinforced concrete specimens of different sizes under three-point bending. Overall speaking, this is a very interesting paper. The numerical method is well explained and the experiments are well designed. The conclusions are also well supported. I think this is paper is ready for publication.
The authors really appreciate the comments of the reviewer and her/his work reviewing the manuscript.

Reviewer 2 Report
Dear Authors,
I appreciate the great effort put into the model part of the research. This is a very important and interesting part of the job. However, I want to make some basic comments about the research part.
1. a thorough literature analysis should be carried out
2. the summary should be corrected as it does not contain a summary, information on tests, methods, samples, basic assumptions. First of all, the basic results of research and analysis were not presented.
3. The research plan and methods raise doubts
- the composition of concrete as the basic material is not shown
- the amount of fibers used may be incorrect and inconsistent with the manufacturer's recommendations. The amount of 10 kg / m3 fibers with a length of 48 mm is not a good solution and raises doubts. Therefore, the amount of cement and the water / cement ratio as well as the aggregates used are of great importance.
- it is advisable to show photographs of the preparation of samples and tests mix concrete and to present all materials, concrete design method and the basis of use of 10 kg / m3 and not e.g. 5 or 6 kg / m3. What was the purpose?
- figure 2, right, shows the false damage to the sample with a fiber content of 10 kg / m3. Such damage proves an incorrectly designed concrete composition. In this case, the amount of cement, whether it is appropriate and the size of the aggregate grain, will matter.
4. The dimensions of the test specimens are not correct. If the element width was 50 mm and the fiber length was 48 mm, the fiber was probably not aligning properly. Was an X-ray examination of the fiber distribution in the cement matrix performed?
5. There is no information on the number of tested samples
6. Notch length: 150 mm raises big doubts? Is this incision correct? According to the standard, it should be 1/10 of the element height?
In my opinion, the study was not conducted properly.
Reviewer
Author Response
Please find enclosed the attached file

Reviewer 3 Report
The technical content of this manuscript is excellent. Results interpretation is clear. Minor comments are listed below.
The allocation of mesh distribution is quite nice. Please provide more details about the simulation parameters which will be useful for others’ reproduction.
It’s unclear the distribution of the fibers.
For figures 8, 10, and 12, please be clear about the meaning of the tricolor bands.
Author Response
Please find enclosed the attached file

Round 2
Reviewer 2 Report
Dear Authors,
Thank You for the answers provided. They are satisfactory for me. I accept the changes.